# Phenotypic and Genetic Spectrum in 309 Consecutive Pediatric Patients with Inherited Retinal Disease

**DOI:** 10.3390/ijms252212259

**Published:** 2024-11-14

**Authors:** Claudia S. Priglinger, Maximilian J. Gerhardt, Siegfried G. Priglinger, Markus Schaumberger, Teresa M. Neuhann, Hanno J. Bolz, Yasmin Mehraein, Guenther Rudolph

**Affiliations:** 1Department of Ophthalmology, University Hospital, Ludwig-Maximilians-University, 80336 Munich, Germany; maximilian.gerhardt@med.uni-muenchen.de (M.J.G.); s.priglinger@med.uni-muenchen.de (S.G.P.); markus.schaumberger@med.uni-muenchen.de (M.S.); guenther.rudolph@med.uni-muenchen.de (G.R.); 2MGZ—Medizinisch Genetisches Zentrum München, 80335 Munich, Germany; teresa.neuhann@mgz-muenchen.de; 3Bioscientia Human Genetics, Institute for Medical Diagnostics GmbH, 55218 Ingelheim, Germany; hanno.bolz@bioscientia.de; 4Institute of Human Genetics, University Hospital, Ludwig-Maximilians-University, 80336 Munich, Germany; yasmin.mehraein@med.uni-goettingen.de; 5Institute of Human Genetics, University Medical Center Göttingen, 37075 Göttingen, Germany

**Keywords:** inherited retinal dystrophy, pediatric patients, syndromic retinal dystrophies

## Abstract

Inherited retinal dystrophies (IRDs) are a common cause of blindness or severe visual impairment in children and may occur with or without systemic associations. The aim of the present study is to describe the phenotypic and genotypic spectrum of IRDs in a pediatric patient cohort in Retrospective single-center cross-sectional analysis. Presenting symptoms, clinical phenotype, and molecular genetic diagnosis were assessed in 309 pediatric patients with suspected IRD. Patients were grouped by age at genetic diagnosis (preschool: 0–6 years, n = 127; schoolchildren: 7–17 years, n = 182). Preschool children most frequently presented with nystagmus (34.5% isolated, 16.4% syndromic), no visual interest (20.9%; 14.5%), or nyctalopia (22.4%; 3.6%; *p* < 0.05); schoolchildren most frequently presented with declining visual acuity (31% isolated, 21.1% syndromic), nyctalopia (10.6%; 13.5%), or high myopia (5.3%; 13.2%). Pathogenic variants were identified in 96 different genes (n = 69 preschool, n = 73 schoolchildren). In the preschool group, 57.4% had isolated and 42.6% had syndromic IRDs, compared to 70.9% and 29.1% in schoolchildren. In the preschool group, 32.4% of the isolated IRDs were related to forms of Leber’s congenital amaurosis (most frequent were *RPE65* (11%) and *CEP290* (8.2%)), 31.5% were related to stationary IRDs, 15.1% were related to macular dystrophies (*ABCA4, BEST1, PRPH2, PROM1*), and 8.2% to rod–cone dystrophies (*RPGR, RPB3, RP2, PDE6A*). All rod–cone dystrophies (RCDs) were subjectively asymptomatic at the time of genetic diagnosis. At schoolage, 41% were attributed to cone-dominated disease (34% *ABCA4*), 10.3% to *BEST1,* and 10.3% to RCDs (*RP2, PRPF3, RPGR; IMPG2, PDE6B, CNGA1, MFRP, RP1*). Ciliopathies were the most common syndromic IRDs (preschool 37%; schoolchildren 45.1%), with variants in *USH2A, CEP290* (5.6% each), *CDH23*, *BBS1*, and *BBS10* (3.7% each) being the most frequent in preschoolers, and *USH2A* (11.7%), *BBS10* (7.8%), *CEP290, CDHR23*, *CLRN1*, and *ICQB1* (3.9% each) being the most frequent in syndromic schoolkids. Vitreoretinal syndromic IRDs accounted for 29.6% (preschool: *COL2A1*, *COL11A1*, *NDP* (5.6% each)) and 23.5% (schoolage: *COL2A1*, *KIF11* (9.8% each)), metabolic IRDs for 9.4% (*OAT, HADHA, MMACHD, PMM2*) and 3.9% (*OAT, HADHA*), mitochondriopathies for 3.7% and 7.8%, and syndromic albinism accounted for 5.6% and 3.9%, respectively. In conclusion we show here that the genotypic spectrum of IRDs and its quantitative distribution not only differs between children and adults but also between children of different age groups, with an almost equal proportion of syndromic and non-syndromic IRDs in early childhood. Ophthalmic screening visits at the preschool and school ages may aid even presymptomatic diagnosis and treatment of potential sight and life-threatening systemic sequelae.

## 1. Introduction

Inherited retinal dystrophies (IRD) affect 1 in 3000 individuals in North America and Europe and are characterized by progressive retinal dysfunction, degeneration, and loss of vision. They represent a frequent cause of blindness in children and adults of working age [1], are highly heterogeneous, and can manifest at any age. IRDs may be isolated or occur as part of multiorgan syndromes and are, with the exception of a few stationary retinal dysfunction disorders (e.g., achromatopsia and congenital stationary light blindness), progressive.

IRDs may be classified according to different clinical criteria based on the primary retinal cell type involved (e.g., rods, cones, retinal pigment epithelium, bipolar, or ganglion cells) or the affected gene. Based on the primarily retinal dysfunction, full-field electroretinography (ffERG) allows for a rough classification of patients into those with macular dystrophy (MD) (normal ffERG), cone dystrophy (CD) (photopic responses reduced only), cone–rod dystrophy (CRD) (cone responses more reduced than rod responses), rod–cone dystrophies (RCD) (rod responses more severely reduced than cone responses, e.g., retinitis pigmentosa (RP)), and some rare electrophysiological subclassifications [2]. Syndromic IRDs are further subclassified according to the type of syndrome with Usher syndromes (USH1 [MIM: 276900], USH2 [MIM: 276901], USH3 [MIM: 276902]), which are characterized by sensorineuronal hearing loss and RP, followed by Bardet–Biedl (BBS [MIM: 209900]) and Alström syndromes (ALMS1 [MIM: 203800]), being the most prevailing subtypes [3]. The terms Leber’s congenital amaurosis (LCA) or early-onset severe retinal dystrophy (EOSRD) are used for subtypes with a particular early onset and severe course of the disease.

Since rhodopsin was identified as the first disease-causing gene in the 1990s [4], more than 290 genes have additionally been identified to cause IRDs (retrieved from: https://retnet.org/summaries#a-genes; RetNet; 23 August 2024), and the number is still increasing. The most prevailing IRD-causing mutations in adult populations are found in *ABCA4* (MIM: 601691), *USH2A* (MIM: 608400), *PRPH2* (MIM: 179605), and *RPGR* (MIM: 312610) [2,5,6,7]. Twenty-seven genes have been attributed to LCA or EOSRD.

Genotype–phenotype correlations are often complex. Different mutations in one gene can cause diverse phenotypes, and mutations in different genes can cause very similar phenotypes that may change with advancing disease. For this reason, the clinical phenotype described in adults may often not correlate with findings at a young age. Clinical diagnosis of IRDs in children is difficult because young children will not report vision loss and may not be diagnosed unless the visual impairment is so severe that it leads to objective symptoms such as nystagmus, photophobia, or light staring in nyctalopic infants. Furthermore, in the early stages of the disease, funduscopy can appear fairly normal, hampering differential diagnosis of stationary disorders or non-retinal origins for visual impairment in young children.

As novel gene-specific therapeutic options are being developed, the identification of individual mutations and early diagnosis has gained importance. Results from clinical trials and experience with the first approved gene therapy, voretigene neparvovec, show that structural preservation appears to be a prerequisite for a beneficial treatment effect [8,9]. Thus, early diagnosis of IRDs is of major importance.

Here, we report results from a cohort of clinically well-characterized children with suspected IRD or syndromic retinal involvement who underwent molecular genetic testing. The results provide insight into the diversity of the mutational and phenotypic spectrum in a childhood IRD cohort that substantially differs from that in adults.

## 2. Results

### 2.1. Cohort Characteristics

Within the inclusion period, a total of 443 patients <18 years old were sent for genetic testing for suspected IRD. Until data analysis, genetic reports were still pending in 101 patients, while 309 returned with a confirmed molecular genetic diagnosis and were included in this analysis; 53.6% were male, and 46.4% were female.

Patients were further grouped according to their age at molecular genetic diagnosis into preschool children aged 0–6 years (n = 127; referred to as the infantile group) and children aged 7–17 years (n = 182; referred to as the juvenile group) and further into isolated and syndromic IRDs in the respective age groups. Median age at molecular genetic diagnosis was 2.5 years (IQR 1.17;4.75) in the infantile group and 11.5 years (IQR 9.25; 14.7) in the juvenile group.

### 2.2. Presenting Symptoms

Information on ocular symptoms at the first visit was available for 122 genetically solved cases in the preschool group (isolated IRD, n = 67; syndromic IRD, n = 55) and for 151 in the juvenile group (isolated IRD, n = 113; syndromic IRD, n = 38). The most frequent reasons for referral in the youngest age group were nystagmus (26.2% [n = 32]), lack of visual interest (18% [n = 22]), nyctalopia (13.9% [n = 17]), declining visual acuity (12.3% [n = 15]), photophobia (9.0% [n = 11]), high refractive errors (15.6% [n = 19]), or a combination thereof. Declining visual acuity (28.5%; [n = 43]), suspected juvenile macular dystrophy (21.2%; [n = 32]), and nyctalopia (11.3%; [n = 17]) were the most common reasons for referral in the 7–17-year-olds (Table 1).

#### 2.2.1. Infantile Cohort

In the younger age group, 34.5% (n = 23/67) of those genetically diagnosed with isolated and 16.4% (n = 9/55) with syndromic forms of IRD were referred for classification of nystagmus. Severe visual impairment from birth (20.9% isolated IRD; 14.5% syndromic IRD), nyctalopia (22.4% isolated IRD; 3.6% syndromic; *p* = 0.007), suspected loss of visual functions (17.9%, isolated; 5.5% syndromic), and photophobia (11.9% isolated; 5.5% syndromic) were key symptoms reported by the parents. In clear contrast to the syndromic preschool group, high hyperopia (isolated 13.4%; syndromic: 0%; *p* = 0.005) was the most common high refractive error in patients with infantile isolated IRD (Table 1).

Out of those children, who later had genetically attributed syndromic IRD, 40.0% (n = 22/55) had been referred by pediatricians for screening of ocular involvement in a suspected syndromic disease. Developmental delay (23.4%); hearing impairment (16.4%); skeletal anomalies (10.9%); kidney anomalies (7.3%); hexadactyly (5.5%); or cardiac anomalies, or microcephaly, or a combination thereof were the most frequent comorbidities. In four syndromic patients, there was no clear evidence for retinal involvement at the time of examination (*MMADHC* (cb1d-MMA), *PMM2* (CDG-1A), *SCO1*, *CTNNB1*). Five patients had obtained a molecular genetic diagnosis of Usher syndrome (*CDH23,* n = 2; *PCDH15,* n = 1; *USH2A,* n = 2) from a genetic screen for early sensorineuronal hearing loss. Clinical examination confirmed features of an early IRD in all five subjects. Four were asymptomatic so far, but in one child, the parents reported night blindness. Overall, 61% had both ocular and extraocular symptoms at the first visit, as disclosed during thorough record-taking. Conversely, in 29% of the preschoolers ophthalmological symptoms were the first signs of a syndromic disease (Table 2).

#### 2.2.2. Juvenile Cohort

Data on initial ocular symptoms were available for 151 genetically solved cases (n = 113 isolated IRDs; n = 38 syndromic IRDs). Declining visual acuity was the key symptom in both isolated (31.0% [n = 43]) and syndromic IRDs (21.1% [n = 8]) in this age group. In agreement with this, suspected macula dystrophy was a major reason for referral (21.2% [n = 32]). After molecular genetic testing, this clinical diagnosis was significantly more commonly attributed to isolated forms of IRDs (*p* = 0.011).

Nyctalopia and nystagmus were represented at comparable frequencies (isolated: 10.6% and 10.6%; syndromic: 13.2% and 10.5%, respectively). High myopia was the most common high refractive error in syndromic IRDs (isolated: 5.3%; syndromic: 13.2%; *p* = 0.179). At the time of referral, 11.9% (n = 18/151) in the total juvenile group reported extraocular features compared to 47.4% (n = 18/38) of those who were then assigned a syndromic genotype. Upon active questioning, however, all syndromic patients for whom this information was available (n = 38/38) reported syndromic features. Data are compiled in Table 1 and Table 2.

### 2.3. Genetic Landscape

#### 2.3.1. Diagnostic Yield and Inheritance Pattern

Molecular genetic diagnosis was available for n = 127 patients in the infantile cohort and n = 182 in the juvenile cohort. Out of these, there were 5.0% (n = 17) in whom only a single heterozygous pathogenic variant was found in combination with a variant of unknown significance (VUS), plus another 5.5% (n = 19) where two VUS were assigned as possibly solved because the phenotype complied with the presumable genotype. No definite molecular diagnosis was established in 10.5% (n = 36), resulting in a diagnostic yield of 78.9% (n = 270/342).

The inheritance mode according to the genes with identified pathogenic variants was autosomal recessive in 65% (n = 198), autosomal dominant in 19% (n = 57), X-linked in 14% (n = 42), and mitochondrial in 2% (n = 6). While the proportion of X-linked inheritance was comparable in the two age groups, that of autosomal dominantly acting variants was almost doubled in the juvenile group (Figure 1).

#### 2.3.2. Disease-Causing Pathogenic Genetic Variants

Pathogenic variants were identified in 96 different genes. The most frequently affected genes with a prevalence of >1% in the whole pediatric cohort were *ABCA4* (15.7%), *BEST1* (5.2%), *CEP290* (4.6%), *CACNA1F* (3.3%), *CNGB3* (3.3%), *COL2A1* (3.3%), *RS1* (3.3%), *USH2A* (3.0%), *RPE65* (2,6%), *TYR* (2,6%), *CHM* (1.9%), *KIF11* (1.9%), *RP2* (1.9%), *ALMS1* (1.3%), *CDH23* (1.3%), and *PROM1* (1.3%). In the 0–6-year-old childhood group, pathogenic variants were found in 61 different genes (n = 35 non-syndromic; n = 26 syndromic), and in the 7–17-year-old group, pathogenic variants were found in 73 (n = 34 non-syndromic, n = 37 syndromic) different genes, accounting for a proportion 52% and 41.2% syndromic genotypes in the 0–6-year-old and 7–17-year-old cohort, respectively. The whole genetic spectrum is illustrated in Appendix A. In the youngest group, 57.4% (n = 73) of the patients were identified as carrying a non-syndromic and 42.6% (n = 55) a syndromic or systemic IRD-causing variant, while in the 7–17-year-old group, 70.9% (n = 129) had non-syndromic and 29.1% (n = 53) had syndromic pathogenic variants, respectively. Figure 2 illustrates the distribution of genes in the two age groups by mode of inheritance. Clinical phenotype and corresponding causative genes are listed in Table 3 and Table 4.

#### 2.3.3. Genes Identified with Pathogenic Variants that Account for Non-Syndromic IRDs

*Infantile cohort.* In the 0–6-year-olds, 32.4% of the isolated retinal dystrophies were attributable to disease-causing mutations in “LCA” genes (*RPE65* (11%), *CEP290* (8.2%), *NMNAT1* (4.1%), *RDH12* and *CRB1* (2.7% each), *RPGRIP1*, *GUCY2D*, and *LRAT* (1.4% each)), and 31.5% were attributed to stationary IRDs. Out of these, 15.1% had mutations in genes associated with forms of albinism, 8.2% had mutations in genes associated with CSNB, and 8.2% had mutations in genes associated with achromatopsia as the most prevalent phenotypes, while 15.1% of the pathogenic variants were associated with early stages of macular dystrophies (*ABCA4, BEST1, PRPH2, PROM1*), and 8.2% were associated with rod–cone dystrophies (*RPGR, RPB3, RP2, PDE6A*). All rod–cone dystrophies were subjectively asymptomatic at the time of genetic diagnosis.

*Juvenile cohort.* In the older children, 41% of the isolated IRDs were attributed to cone-dominated disease (macular dystrophies or cone–rod dystrophies). Mutations in the *ABCA4* gene alone accounted for 34.1% of isolated IRDs, 10.3% were found in *BEST1*, whereas mutant alleles accounting for non-syndromic rod–cone dystrophies were found in 10.3% of the 7–17-year-old group (*RP2, PRPF3, RPGR; IMPG2, PDE6B, CNGA1, MFRP, RP1*). Stationary IRDs constituted 14.2% (3.1% albinism, 4.7% CSNB, 5.4% achromatopsia), variants causative for LCA were assigned to 8.5% (*CEP290, GUCY2D, RDH12, CREB1, RPE65, RD3*), 6% were annotated in *RS1*, and 3.1% were annotated in *CHM* (Table 3, Appendix A).

#### 2.3.4. Genes Identified with Pathogenic Variants that Account for Syndromic IRDs

*Infantile cohort.* A total of 37% had mutations causative for ciliopathies (11.1% Usher syndrome: *USH2A* (5.6%), *CDH23* (3.7%), *PCDH15* (1.9%); 9.3% Joubert or Senior Loken syndromes:: *CEP290* (5.6%), *IQCB1* (1.9%), *C5orf42* (1.9%); 9.3% Bardet–Biedl syndrome: *BBS1* (3.7%), *BBS10* (3.7%), *BBS2* (1.9%)), while 29.6% had vitreoretinal syndromic disorders (including causative mutations in *CTNNB1, KIF11, LAMA1, NDP, PACS2, RNU4ATAC*). Among these, Stickler syndromes (14.8%: *COL2A1* (5.6%), *COL2A1*-associated Kniest dysplasia (3.7%), *COL11A1* (5.6%)) were the most prevalent entities, underlying metabolic diseases were found in 9.4% (*OAT, HADHA, MMACHD, PMM2*), 5.3% had syndromic features of phenotypic albinism (*HPS5, HPS6, LYS*), and 3.8% were mitochondriopathies. Four patients had chromosomal microdeletions associated with an IRD.

*Juvenile cohort*. In the 7–17-year-olds, 45.1% of the syndromic patients carried disease-causing mutations in ciliopathy-associated genes (19% Usher syndrome: *USH2A* (11.7%); *CDH23* (3.9%), *CLRN1* (3.9%); 11.8% Joubert or Senior Loken syndromes: *CEP290* (3.9%), *CEP120* (1.9%); *IQCB1* (3.9%), *WDR19* (1.9%); 9.8% Bardet–Biedl syndrome: *BBS10* (7.8%); *BBS12* (1.9%). Comparable to the infantile group, vitreoretinal syndromic IRDs accounted for the second largest group (23.5%), with 11.6% carrying Stickler genotypes (*COL2A1* (9.7%), *COL18A* (1.9%)), followed by 9.8% KIF11-associated IRD, and 7.8% were identified as mitochondriopathies (KSS (3.9%), *MT-TL1* (1.9%), Pearson syndrome (1.9%)). Two patients had chromosomal microdeletions, and one had a microduplication. Genes, number of patients, and clinical phenotype are compiled in Table 4 and Appendix A. Figure 3 illustrates the diversity of the genetic landscape of the syndromic IRDS in the present study.

Genes implicated in the infantile group but not in the juvenile cohort, as well as overlapping genotypes, are listed separately in Appendix A.

### 2.4. Revision of Initial Clinical Diagnosis

Prior to genetic testing, patients had been assigned clinical diagnoses by clinical phenotype in isolated IRDs (e.g., LCA, MD, RCD, achromatopsia, Best’s disease). Syndromic IRDs were categorized by the specific subtype of syndrome as outlined in Table 5. Molecular genetic testing then allowed for more precise subclassification and refinement of the clinical diagnoses as follows.

#### 2.4.1. Infantile Group

In the preschool cohort, 25.2% (32/127) had been clinically classified as LCA. Out of these, 75% (24/32) had isolated, and 21.8% (7/32) had syndromic genotypes. In one patient, the clinical diagnosis of LCA had to be refined to X-linked retinoschisis after molecular genetic testing. The clinical phenotype of albinism (12.6% (16/127)) was genetically confirmed in 81.3% (13/16), while 18.8% (3/16) were identified as Hermansky–Pudlak or Chediak–Higashi syndromes. The clinical diagnosis of achromatopsia (6.3% (8/127)) correlated with the genotype in 75% (n = 6), but in 25% (n = 2), pathogenic variants accounting for Alström syndrome were found. Specific clinical phenotypes (Stargardt’s disease, choroideremia, Best’s disease, and Bardet–Biedl and Stickler syndromes) correlated well with the disease-causing molecular genetic spectrum. Rod–cone dystrophies (RCD) and RP were suspected in 6.3 % (n = 8/127). Remarkably, half of them (n = 4/8) were genotypically attributed to syndromic IRDs (syndromic: *COH1, USH2A, RNU4ATAC*; Pearson syndrome; isolated: *RPGR* (2), *RP2, CACNA1F*). Of note, phenotypic macular dystrophy other than Stargardt’s or Best’s disease was assigned to only 2.4% of the whole cohort, but all were then found to have a syndromic genetic background. Furthermore, in the youngest age group with isolated IRD, six children were clinically asymptomatic but diagnosed with an early IRD by clinical examination and retinal imaging in a screening visit because of a positive family history for an IRD (*ABCA4*, n = 1; *RPGR*, n = 1; *CHM*, n = 1; *BEST1*, n = 1; *PRPH2*, n = 1; *PDE6A*, n = 1). Two siblings had undergone WES for skeletal anomalies in the older child, which evidenced a likely pathogenic frame-shift variant together with a VUS in a highly conserved domain in the *RBP3* gene in the compound heterozygous state. Until October 2023, both were clinically asymptomatic, but in the older group, by then 10-year-old children, peripheral RPE mottling was arising.

#### 2.4.2. Juvenile Cohort, 7–17 Years

In this age group, 20.4% (35/172) were clinically classified as retinitis pigmentosa or RCD, and 10.5% (18/172) were clinically classified as CRD. Molecular genetic testing then refined 60% (21/35) of the RCDs as isolated IRD and 40% as syndromic forms (Table 5), while 94.4% (17/18) of the CRDs were related to mutant alleles causing non-syndromic CRD.

Bardet–Biedl and Stickler syndromes were the most common specific, clinically diagnosed syndromes (12.8% of the syndromic phenotypes each). In BBS patients, Stickler syndromes, CHM, and Stargardt’s disease, the clinically suspected diagnosis correlated well with the genetic diagnosis. The clinical phenotypes of achromatopsia and macular dystrophy (clinical diagnoses of Best’s disease and Stargardt’s excluded) were equally represented in isolated and syndromic cases (Table 5). Macula dystrophies phenotypically different from Stargardt’s were suspected in 7.6% (13/172), and 30% (4/13) of these turned out to have a syndromic genetic background (*CLN3* (2), *CDH3* (2)). A presymptomatic clinical diagnosis prior to the onset of symptoms was made in three patients with a positive family history (*ABCA4*, n = 1; *CHM*, n = 1; *IRXB*-Cluster, n = 1).

## 3. Discussion

In the present study, we analyzed the genetic and phenotypic spectrum of 309 consecutive pediatric patients diagnosed with IRDs at a median age of 7.4 years (IQR 3.5;12.4) at molecular genetic diagnosis. This is one of the youngest and largest pediatric patient cohorts worldwide, as far as the authors are aware.

In order to specify those genotypes that we can detect the earliest, we further subclassified subjects six years old or younger at diagnosis (referred to as the infantile group) and compared the genetic and phenotypic spectrum to patients 7 to 17 years old at the time of diagnosis (referred to as the juvenile group). Three main findings emerge from this. First, the genetic and phenotypic spectrum differs markedly not only between children and adults but also between children of different age groups. Second, the proportion of syndromic forms, whose comorbidity does not necessarily have to be severe, is far higher than in adults. Third, genetic screening is initiated because of comorbidities, and modern ophthalmological imaging techniques can detect presymptomatic IRDs. To discuss our findings within the context of the published literature, we chose to refer to three recent studies on pediatric cohorts [10,11,12]. These studies included patients younger than 18 years but did not subclassify infantile and juvenile patients. For comparability, we will, therefore, refer to the whole cohort in some aspect.

Inheritance modes in our whole pediatric cohort differed markedly from two British cohorts in the frequency of X-linked (20.7%) [10] and autosomal dominantly acting variants (7.6%) [11]. In addition to age- and population-related bias, an explanatory reason for the lower frequency of autosomal dominant variants in Taylor’s pediatric cohort may be that the data stem from a multipanel analysis from 2014 to 2016 [11]. Since then, panels and sequencing technologies have repeatedly been optimized, and detection rates have continuously been increasing. In particular, the diagnostic rate for genes associated with autosomal dominant inheritance in 2015 was reportedly 10 times lower than for autosomal recessively inherited ones [2,13] because candidate homo- or heterozygous biallelic variants can more easily be detected by homozygosity mapping in consanguineous families and because of their low prevalence in an individual, whereas the monoallelic, potentially dominantly acting novel variants can be difficult to identify in small families [2,12]. Furthermore, targeted sequencing in patients with pathognomonic clinical or ERG phenotypes, such as *CHM, RS1, BEST1*, or C*ACNA1F*, as well as specific sequencing of *ORF15* in *RPGR,* may contribute to differences between studies. For example, *CACNA1F* was the most commonly mutated allele (14.9%) in the Manchester pediatric study but consisted of only 2.6% of our whole pediatric cohort. Of note, in our cohorts, the proportion of autosomal dominant variants was almost doubled in juvenile (24%) versus infantile patients (13%), almost reaching prevalence (23%) in a mainly adult German cohort with a comparable catchment area [2]. This fits well with the traditional assumption that autosomal dominant IRDs appear later in life [14,15,16].

X-linked traits were comparable in both age groups, albeit with different distributions. *CACNA1F* represented the most abundant X-linked genotype in the infantile subgroup (21%) and was in third place in the juvenile X-linked IRDs (17%). Interestingly, initial clinical symptoms of the main X-linked conditions related to *CACNA1F, CHM, RPGR,* and *RP2* differed considerably. All infantile individuals with *CHM* were brought by parents for screening due to a positive family history. Night blindness was not reported, but all had first funduscopic and obvious autofluorescence signs of choroideremia. In the 7–17-year-old group, night blindness was solely reported by the parents but was not perceived as personally restricting by the patients. There were no marked refractive errors. Infantile subjects with *CACNA1F* mutations presented with nystagmus (n = 3), closing-up when reading (n = 1), and suspected buphthalmos (n = 1). All were myopic. Likewise, in the older age group, high myopia and/or nystagmus in early childhood were reasons for referral and consecutive genetic testing, but none complained of night blindness at that time. This agrees with previous reports demonstrating that 40% of the *CACNA1F* patients may not have night vision problems. For this reason, others have already been questioned on whether the term “congenital stationary night blindness” is entirely appropriate for *CACNA1F*-associated disease [13]. This was in clear contrast to those with *RPGR* and *RP2* variants, with 81% who were myopic and suffered from more severe night blindness in the juvenile cohort. Like those with *CACNA1F* variants, infantile *RPGR* patients all were referred for high myopia and/or nystagmus but not for night blindness, whereas the juveniles self-reported marked night blindness upon questioning, and in those with pathogenic *RP2* variants, night blindness was a cardinal and restricting symptom. This is in line with the findings from Prokofyeva et al., who found that patients with X-linked RP experience night blindness at a median age of 16 [16]. In view of this, in young patients with high myopia and ERG anomalies, *CACNAF1, RPGR*, and *RP2* genotypes should be excluded even in the absence of night blindness.

Considering high myopia as a presenting symptom in children, another finding from this study is that Stickler syndromes are a valuable differential diagnosis in pediatric patients of both age groups. These constituted the most common autosomal dominant conditions in 0–6-year-olds (45%) and accounted for 12% of the juvenile cohort. Stickler syndromes are among the most frequently inherited connective tissue disorders and are the most common cause of inherited and childhood retinal detachment [13,17,18]. Exemplary for this, two patients in our infantile cohort diagnosed with Stickler type I were referred for retinal detachment. Stickler syndromes should be suspected if high myopia from birth coincides with often only mild facial dysmorphia, palatine clefts, or sensorineural hearing impairment. This could be of therapeutic relevance since in type 1 Stickler syndrome, prophylactic 360° cryotherapy or laser photocoagulation may reduce the risk of giant retinal tears [17]. Although this approach is controversially discussed, and it is difficult to explain to parents that a ‘healthy’ eye should be exposed to cryotherapy or laser photocoagulation, this should be considered and offered in view of the study situation.

In adult IRD cohorts, phenotypic RP is reported to account for 43–49% of cases [2,6,7,10,19]. In clear contrast, in our infantile cohort, most cases were phenotypically and genetically attributed to LCA (25.2%, n = 32), followed by albinism (12.7%, n = 16), while the frequency of rod–cone phenotypes was only 6.3% (n = 8). In the juvenile group, the phenotypic spectrum started to approach that in adults [2,7,10,20]: RCDs and RP together constituted 20.4% (n = 35), followed by Stargardt’s disease (11.6%, n = 20), CRD (10.5%, n= 18), Best’s disease, LCA, and other forms of MD (7.6%, n = 13, each). For comparison, in a large, predominantly adult southwestern German cohort, the respective proportions were 47%, 5.9%, 8.2%, and 8.8% [2].

Variants in the 28 genes that were most commonly affected were responsible for 69.9% of the genetically solved cases in the whole cohort. The top 10 constituted five autosomal recessive (*ABCA4, CEP290, CNGB3, RPE65, USH2A*), two autosomal dominant (*BEST1, COL2A1*), and two X-linked conditions (*CACNA1F, RS1*). While the most common genetic diagnoses in infantile patients (*RPE65, CEP290, ABCA4, BEST1, CACNAF1, CNGB3, TYR, COL2A1, COL11A1*) were rather evenly distributed with frequencies from 3% to 6%, the relative abundance of *RPE65* variants (5.8%) may be biased from the specification our center on gene therapy. *ABCA4*-related disorders represented the most abundant by far and, in the juvenile cohort, even reached a frequency of 24%. This is even slightly higher than in adult populations, which range from 10.4% to 20.8% [2,7,10,21]. This is comparable to the high frequency of the clinical diagnosis of Stargardt’s disease (11.6%) and non-syndromic CRDs in this age group (10.5%).

In other European pediatric patient populations, the spectrum of the most common disease-causing genes is largely comparable, although, as mentioned above, individual abundances differ [5,10,11]. Interestingly, *CEP290* (consisting of both syndromic and non-syndromic cases) was the third most frequent genotype in our overall cohort but was not referred to as frequently in the, e.g., Emirati or Manchester British pediatric cohorts. The reason for this remains uncertain since all infantile cases phenotypically presented as LCA, three in the juvenile cohort complained of either photophobia or declining visual acuity or nyctalopia, and all subjects had obvious funduscopic or electrophysiological signs of an IRD, which would prompt molecular genetic testing if available. The largest deviations from the European genotypic spectrum were observed in a pediatric cohort from the Arabian Gulf, where Khan et al. found *KCNV2, CRB1, CNGA3, MERTK*, and *IMPG3* amongst the most frequent causative genes. As stated by the author, high rates of consanguinity in the studied population may account for these differences. This is further underscored by the high prevalence of homozygous autosomal recessive variants and recurrent mutated alleles in *MERTK* and *KCNV2*, which are highly suggestive of a regional founder effect [12].

The most striking observation in this study, however, was the high diversity and frequency of syndromic conditions in the pediatric cohort, with 52% in the infantile (32 different genes) and 41.2% in the juveniles (25 different genes) accounting for syndromic or systemic diseases. When considering the number of affected patients, 42.5% of infantile cases and 29.1% of juvenile patients were syndromic, while the number of different genotypes was comparable to that of younger children. This is remarkable when compared to adult cohorts, where 20–30% of RP are estimated to be syndromic, with Usher syndrome being the most prevailing phenotype by far. Also, the genetic landscape in adult cohorts does not appear as diverse as that of this pediatric cohort. For instance, in a recent retrospective cohort study of syndromic RP in Portugal, Cortinhal et al. found causative variants in 25 genes, and Usher syndromes reached a prevalence of 62% [22]. In our infantile group, only 10% were attributed to Usher syndromes, all of which were identified in a screening for sensorineural hearing before they developed night blindness, while this number increased to 19% in the juvenile group. This highlights nicely that the retinopathy in Usher syndrome has a later onset.

While in the infantile group, extraocular symptoms often were striking and raised suspicions of a syndromic condition, they may be only mild in older patients and primarily not seen as part of a systemic condition. Particularly in older children, polydactyly surgically corrected in early childhood can easily be overlooked, speech delays or hearing impairments have long been accepted as isolated, and truncal obesity as an issue is often negated by the parents.

Most importantly, however, our data suggest that the possibility of a systemic association in children must be considered in all main clinical phenotypes at any age. In both age groups, at first sight, purely ocular symptoms such as delayed visual maturation, photophobia, high refractive errors, declining visual acuity, or night blindness in juvenile patients can be part of a syndromic condition, where extraocular comorbidities may manifest later. For example, 22% of 0–6-year-olds and 25% of 7–17-year-olds who were primarily classified as LCA carried causative mutations in genes associated with syndromic retinopathies. Out of those with phenotypic albinism, 18.7% in the infantile group and 40% in the juvenile cohort were genetically assigned to syndromic forms, and in both age groups, 25% of those classified as achromatopsia were found to have pathogenic variants causative of Alström syndrome. One patient, who was referred at the age of 6 for an evaluation of blindness, had suffered a cardiac arrest in the first days of life due to suspected CMV myocarditis. The child was resuscitated and developed a pendular horizontal nystagmus in the first weeks of life, which since then was attributed to a cerebral visual impairment in the context of a suspected hypoxic brain injury. The child was markedly photophobic, fundus autofluorescence revealed a concentric ring of increased autofluorescence in the middle periphery, and the light-adapted 30 Hz flicker response was unrecordable. Molecular genetic testing then confirmed a diagnosis of Alström syndrome. The heart failure was thus attributed to *ALMS1*-associated cardiomyopathy. For several of these children, molecular genetic diagnosis can facilitate timely and preventive access to treatment of their comorbidities.

There are several limitations to this study inherent to its retrospective nature. The date of first symptoms, clinical diagnosis, and family history were not available for all patients. For this reason, the mode of inheritance was retrospectively assigned according to the identified variants, and data on clinical phenotypes were assessed for a subset of the patients. Furthermore, a strength but at the same time confounding factor may derive from a referral bias. Our ocular genetics service is part of the Department of Pediatric Ophthalmology and Strabism at LMU Munich and is in close collaboration with the Dr. v. Hauner’sche Pediatric Hospital, which is the largest institution for sick children in Germany. From there, many children with systemic disease are referred for screening of an ocular involvement, but not all of them are seen by the authors of this study and, therefore, may not be filed in the database. From this, it must be assumed that the proportion of children with syndromic or systemic IRDs is even higher. Another limitation is that the number of cases with albinism, Best’s disease, and Stickler syndromes may be underrepresented in this cohort because, until 2021, diagnosis in these entities relied on clinical assessment. Also, some of those in the juvenile cohort classified with causative genes for syndromes that manifest early in life, such as, for example, Senior Loken, Joubert, or Alström syndrome, may have obtained a delayed molecular diagnosis because genetic testing was not carried out or successful before 2016. The same applies to children with isolated nystagmus, in whom molecular genetic diagnosis may have been initiated later when subjects initially diagnosed with idiopathic infantile nystagmus in the pre-imaging area later developed signs of visual impairment and IRDs.

## 4. Materials and Methods

### 4.1. Patient Recruitment and Phenotypic Data Collection

This single-center retrospective cross-sectional study included patients seen in the retinal genetic and pediatric ophthalmology outpatient clinic at the Department of Ophthalmology of the Ludwig Maximilian University of Munich from November 2016 to October 2023. The retinal genetic and pediatric ophthalmology clinic is a secondary and tertiary referral center where patients are referred when their primary care physician or ophthalmologist or the Institutional Children’s Hospital suspects an IRD. All patients were seen by three experienced clinicians (CSP, MG, GR). Inclusion criteria included the following: age of 17 years or younger at genetic testing; a clinical diagnosis of an inherited retinal disease by one of the senior clinicians (CSP, GR) based on the patient’s history of visual symptoms, clinical examination, retinal imaging, and—when available—electrophysiology and Goldman perimetry; and that the patient underwent genetic testing for confirmation of the clinical diagnosis in the inclusion period. Once a case was considered genetically solved, age of onset, clinical phenotype, symptoms at onset, visual acuity at first visit, refraction, and causative genotypes were recorded in an internal database on the LMU server. The search end date was 25 October 2023.

### 4.2. Clinical Examination and Genetic Testing Pathway

Full medical, ocular, and developmental history was assessed for each patient. Information on age of symptom onset was assessed as reported by the patients or parents in case of early-onset IRD or as diagnosed by the referring ophthalmologist at first visit. When extraocular features were suspected, children were referred to the Dr. v. Hauner’s Children’s Hospital, LMU Munich, for further work. Ophthalmological examination included age-appropriate assessment of best-corrected visual acuity, slit lamp examination, dilated fundus examination, orthoptic assessment, cycloplegic refraction, and retinal imaging, including spectral domain optical coherence tomography (OCT), fundus autofluorescence (AF) imaging (Spectralis HRA + OCT™, Heidelberg Engineering, Heidelberg, Germany), color fundus photography (Clarus™, Zeiss, Oberkochen, Germany), or wide-field color fundus imaging and autofluorescence (Optos California™, Optos, Düsseldorf, Germany). Goldmann perimetry and full-field ERG were performed in most cases. The protocols adhered to the International Society of Electrophysiology of Vision (ISCEV) standards [23]. In non-cooperative younger children, a reduced protocol with photopic 30 Hz flicker was used to confirm retinal dysfunction. The decision on which examination was performed was made individually. If patients were suspected to have an IRD, genetic testing was suggested and discussed. In Germany, genetic testing is covered by public health insurance but not always by private health insurance. For this reason, genetic sampling was offered to those patients and families with suggestive IRDs who had access to public health insurance and were interested in being tested. Those with private health insurance could solely be tested to see if the insurance company agreed to cover the costs or if the patient paid the costs themself. A flow chart illustrating the clinical pathway of the study subjects is provided in Figure 4.

### 4.3. Molecular Genetic Analysis and Classification of Cases and Variants

Genetic counseling and initiation of genetic testing were carried out by the Institute of Human Genetics of the Ludwig Maximilian University or the Medical Genetics Center (MGZ) in Munich. Molecular genetic testing applying the multiple gene panel approach was performed at three sites: CeGaT GmbH/Tübingen, MGZ (Medizinisch Genetisches Zentrum/Munich), and Bioscientia Human Genetics/Ingelheim, all of which are accredited by DAkkS according to DIN EN ISO 15189 (for details see: https://www.dakks.de/de/medizinische-labore-din-en-iso-15189.html, 23 August 2024). In the inclusion period, from 2016 to 2019, genetic services mainly relied on NGS of gene panels. Since 2019, NGS whole-exome sequencing (WES) has been the most frequently applied method. Virtual panels were bioinformatically extracted based on human phenotype ontology (HPO) data for core phenotyping. The virtual panels in all three centers were constantly adapted in agreement with the latest literature as new genes were discovered. Ultimately, using this approach, all candidate genes known at that time were read out.

In the most comprehensive NGS panels for autosomal recessive or sporadic cone–rod or rod–cone dystrophies, up to 214 genes were offered. With the aim of increasing sensitivity, clinicians narrowed down the requested gene analysis to subpanels based on the clinical phenotype and presumable mode of inheritance from pedigree analysis. For cases with clinical diagnosis of RP, subpanels comprised either autosomal recessive and X chromosomal inherited genes or autosomal dominant and X chromosomal inherited genes. For macular dystrophies suggestive of Stargardt’s or Best’s disease, restricted panels were requested. Single gene analysis was performed in patients with specific clinical phenotypes such as *RS1* or *CHM*. In all IRD panels, the ORF15 exon of RPGR was always covered. Likewise, clinical and electrophysiological features of albinism, achromatopsia, stationary night blindness, LCA, cone or cone/rod dystrophy prompted the respective panels, as did patients with presumable ciliopathies or dysmorphic features combined with high myopia without night blindness suggestive of Stickler syndromes.

In the respective genes, only variants in the coding region and the flanking intronic regions with a minor allele frequency (MAF) < 1.5% were evaluated. Variants found by NGS were classified and interpreted according to the American College of Medical Genetics and Genomics (ACMG) guidelines employing information from the Human Gene Mutation Database (HGMD), ClinVar Database, the Genome Aggregation Database (gnomAD; https://gnomad.broadinstitute.org, 23 August 2024), and Single-Nucleotide Polymorphism Database (retrieved from: https://www.ncbi.nlm.nih.gov/snp, 23 August 2024), as well as additional variant prediction programs (e.g., REVEL and other) allowing for grouping into pathogenic, likely pathogenic, unknown significance, and likely benign and benign variants. Population frequencies were taken from the Genome Aggregation Database (gnomAD; retrieved from: https://gnomad.broadinstitute.org, 23 August 2024) and Single-Nucleotide Polymorphism Database (retrieved from: https://www.ncbi.nlm.nih.gov/snp, 23 August 2024) with continuous updates during the inclusion period. Applying this classification system, a patient case was considered genetically solved if the patient (1.) was heterozygous for two variants classified as “pathogenic” or “likely pathogenic” in a gene known to cause autosomal recessive disease, (2.) heterozygous for a variant considered “pathogenic” or “likely pathogenic” in a gene attributed to autosomal dominant disease, (3.) harbored “pathogenic” or “likely pathogenic” variant in a gene associated with X-linked disease—hemizygous in a male in an X-linked recessive condition or heterozygous in female in an X-linked dominant condition, or (4.) mitochondrial if a “pathogenic” or “likely pathogenic” variant was found in genes known to cause a mitochondriopathy. (5.) In case of a recessive disease and identification of a VUS, a patient was considered “possibly solved” if they carried the VUS on one allele and a pathogenic/likely pathogenic variant on the other and if the genotype agreed with the clinical phenotype. (6.) Cases where only a heterozygous pathogenic variant for an a.r. disease was found were considered as possibly solved when compatible with the clinical phenotype. Segregation analysis in parents or close relatives was performed when possible. Mode of inheritance in data assessment was classified based on results from genetic testing, both in sporadic and familial cases.

### 4.4. Statistical Analysis

The software SPSS (version 29.0.2.0; IBM, Armonk, NY, USA) was used for statistical analysis. Descriptive statistics were calculated for all variables. A *p*-value < 0.05, as computed by two-sided Fisher’s exact test, was defined as statistically significant.

### 4.5. Consent and Ethical Approval

All patients gave written informed consent for genetic testing. For subjects who were underage at the time of blood sampling, informed consent was obtained from the parents or legal guardians. This study was approved by the Institutional Review Board of the Ethics Committee of the Ludwig Maximilian University of Munich under study number 22-0897 and adhered to the tenets of the Declaration of Helsinki.

## 5. Conclusions

In the present study, we, for the first time, provide an overview of the genetic and phenotypic landscape in 309 pediatric patients in Germany suffering or suspected of becoming affected by some form of IRD in the future. This is exemplary of the recent advances in genetic profiling, enabling ophthalmology to take a deep dive into precision medicine with an impact not only on disease prognosis but also on treatment prospects and genetic counseling for family members. This is beneficial for the affected in many aspects: Treatment has become a real-life scenario for *RPE65*-associated IRD since 2017; *CEP290, CNGB3, RS1, CHM,* and *RPGR* have been or are the subject of gene therapy trials [24,25,26,27,28,29,30]; and results from pediatric patients treated so far strongly suggest that treatment early in the disease progress is key for the functional outcome [8,9]. Furthermore, early identification of the underlying genotype has prognostic value, not only with respect to a stationary or progressive disease but, even more importantly, to rule out a treatable systemic condition.

Nevertheless, there may also be a downside to these advances. As shown here, IRDs can be identified by molecular genetic testing or routinely used diagnostic imaging techniques such as fundus autofluorescence, OCT, and ERG in early childhood before symptom onset. Such deep clinical phenotyping in asymptomatic patients can be of particular relevance for subjects with a.d. or x-linked pedigrees as well as in younger siblings of children with autosomal recessive traits and may encourage clinicians to proceed to molecular genetic testing. In this respect, local regulations must be carefully considered. In Germany, for example, the Genetic Diagnostics Act (§ 3 No. 8 GenDG) stipulates that predictive molecular genetic testing must not be performed on presymptomatic patients who are unable to give informed consent.

In view of this, the ethical question arises on whether OCT, with AF as a prerequisite for initiating molecular diagnostics, should be offered as part of pediatric ophthalmological screening or rather only if effective treatment options become available in order to enable presymptomatic treatment. Would this knowledge, therefore, rather be a severe psychological stress for child and family years before the onset of symptoms, or do advantages far outweigh this putative burden, as for several of the syndromic children, a thoughtful ophthalmological examination may aid in early molecular genetic diagnosis of their syndromes and facilitate preventive care of the comorbidities associated with the identified genes.

## Figures and Tables

**Figure 1 ijms-25-12259-f001:**
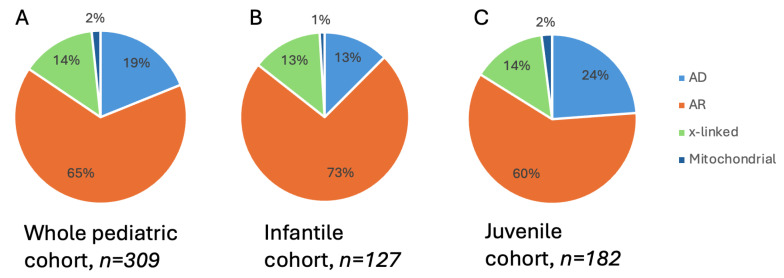
Modes of inheritance for inherited retinal dystrophies in a molecularly genetically characterized pediatric cohort. Pie charts show the subsets of individuals with diseases associated with X-linked, autosomal dominant (AD), autosomal recessive (AR), and mitochondrial variants. (**A**) The left panel shows the subsets for the whole pediatric cohort. (**B**) The middle panel is for the subset of the overall cohort with affected individuals under 6 years. (**C**) The right-hand panel shows the subset of individuals 7–17 years old at time of genetic diagnosis.

**Figure 2 ijms-25-12259-f002:**
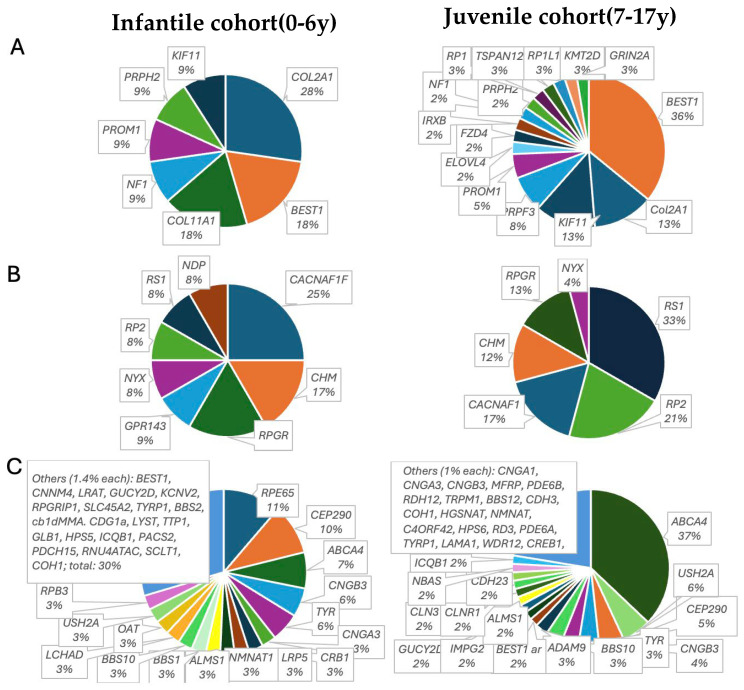
Classification of mutated genes in genetically solved pediatric patients with isolated or syndromic IRDs. (**A**) shows the distribution of autosomal dominant, (**B**) X-linked, and (**C**) autosomal recessive variants. Left panels: infantile cohort, 0–6 years old at time of molecular diagnosis. Right panels: juvenile patients, 7–17 years old at time of molecular diagnosis. The percentage of mutation of each gene is given below the gene.

**Figure 3 ijms-25-12259-f003:**
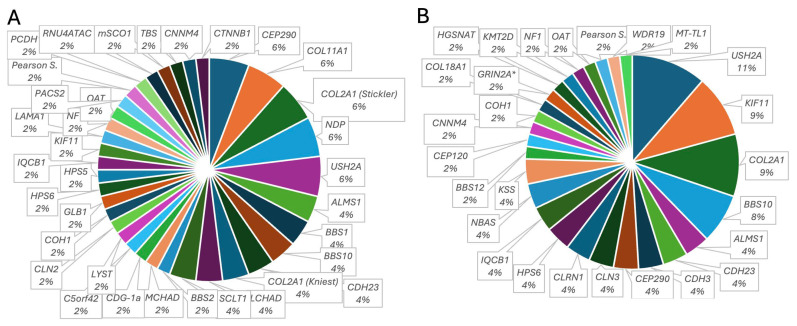
Diversity of genes carrying causative pathogenic variants in a cohort of pediatric patients with genetically solved syndromic IRD. Mutated genes identified in (**A**) patients 0–6 years old and (**B**) 7–17 years old at molecular genetic diagnosis. ***, presented with nystagmus, suspected maculopathy, papilledema, and optic disc drusen. The percentage of mutation of each gene is given below the gene.

**Figure 4 ijms-25-12259-f004:**
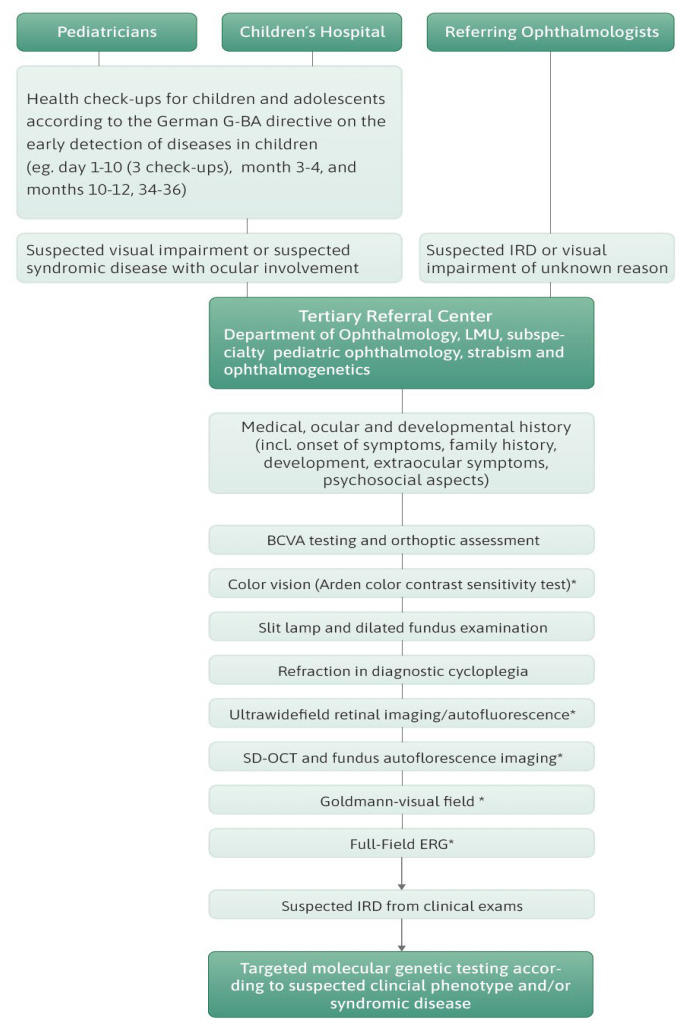
Flow chart illustrating the clinical diagnostic pathway of the patients in the present study.

**Table 1 ijms-25-12259-t001:** Ocular symptoms and reasons for referral at first visit.

	Infantile Cohort (0–6 y)						Juvenile Cohort (7–17 y)					
	Whole Infantile Cohort		Isolated		Syndromic		Whole Juvenile Cohort		Isolated		Syndromic	
	%^+^	n	%	n	%	n	%^+^	n	%	n	%	n
Nystagmus	26.2	32	34.3	23	16.4	9	10.6	16	10.6	12	10.5	4
No visual interest	18.0	22	20.9	14	14.5	8	4.6	7	2.7	3	10.5	4
Nyctalopia	13.9	17	22.4	15	3.6 **	2	11.3	17	10.6	12	13.2	5
Declining VA	12.3	15	17.9	12	5.5	3	28.5	43	31.0	35	21.1	8
Photophobia	9.0	11	11.9	8	5.5	3	3.3	5	4.4	5	0	0
High myopia	8.2	10	6.0	4	10.9	6	7.3	11	5.3	6	13.2	5
High hyperopia	7.4	9	13.4	9	0 **	0	1	2	1,8	2	0	0
Strabism	7.4	9	9.0	6	5.5	3	4,0	6	4.4	5	2.6	1
Suspected IRD	7.4	9	3.0	2	12.7	7	9.9	15	10.6	12	7.9	3
Suspected MD	1.6	2	3.0	2	0	0	21.2	32	25.7	29	7.9 *	3
Visual field defects	2.5	3	3.0	2	1.8	1	3,3	5	1.8	2	1.8	1
Suspected Chorioretinitis	0	0	0	0	0	0	2.6	4	2.7	3	2.6	1
Hamartoma, retinal detachment, and others	7.4	9	6.0	4	9.0	5	3.3	5	3.5	4	2.6	1

VA, visual acuity; IRD, inherited retinal dystrophy; MD, macular dystrophy; %^+^, percentage of cases of the whole infantile cohort; %, percentage of cases of the respective genetic background (variants attributed to isolated or syndromic IRDs, respectively); n, number of cases for whom information reasons and/or ocular symptoms upon referral was available; * *p* < 0.05; ** *p* < 0.01, Fisher’s exact test.

**Table 2 ijms-25-12259-t002:** Extraocular symptoms at first referral in pediatric syndromic IRD.

	Infantile Cohort (0–6 y)		Juvenile Cohort (7–17 y)	
	% of Cases	n Cases	% of Cases	n Cases
Developmental delay	23.4	13	15.8	6
Hearing loss	16.4	9	21.1	8
Skeletal anomalies	7.3	4	5.3	2
Kidney anomalies	7.3	4	5.3	2
Polydactyly	5.5	3	23.7	9
Cardiac anomalies	5.5	3	7.9	3
Microcephaly	5.5	3	5.3	2
Cerebral anomalies (other)	3.6	2	2.6	1
Palatine cleft	3.6	2	5.3	2
Ptosis	1.8	1	5.3	2
Epilepsia	1.8	1	2.6	1
Obesity	0	0	1.8	1

**Table 3 ijms-25-12259-t003:** Clinical diagnoses and distribution of causative mutations in genes attributed to isolated IRDs in infantile (0–6 years old) and juvenile (7–17 years old) at time of genetic diagnosis.

	Age Group: 0–6 Years				Age Group: 7–17 Years	
Clinical Diagnosis	%	n	Associated Genes (n Cases)	%	n	Associated Genes (n Cases)
LCA	32.4	24	*RPE65 (8), CEP290 (6), NMNAT (3), CRB1 (2), RDH12 (2), LRAT, GUCY2D, RPGRIP1 (1)*	8.5	11	*CEP290 (4), GUCY2D (2), RDH12 (2), CREB1, RD3, RPE65*
Albinism	14.9	11	*TYR (4), GPR143 (3), OCA2 (2), SLC45A2, TYRP1 (1)*	3.1	4	*TYR (3), TRPM1*
MD and Cone–Rod Dystrophy	10.8	8	*ABCA4 (6), PRPH2, PROM1 (1)*	41.3	52	*ABCA4 (43), ADAM9 (3) PROM1 (3), PRPH2, RP1L1, ELVOLF4*
Achromatopsia	8.2	6	*CNGB3 (4), CNGA3 (2)*	5.4	7	*CNGB3 (6), CNGA3 (1)*
CSNB	8.2	6	*CACNAF1 (5), NYX (1)*	4.6	6	*CACNAF1 (5), NYX (1)*
Rod–Cone Dystrophy	8.2	6	*RPB3 (2), RPGR (2), RP2 (1), PDE6A (1),*	10.3	13	*RP2 (5), PRPF3 (3), RPGR (3), IMPG2 (2), PDE6B (2), CNGA1, MFRP, RP1*
Best’s Disease	4.1	3	*BEST1 (3)*	10.3	13	*BEST1 (13)*
CHM	2.7	2	*CHM (2)*	3.1	4	*CHM (4)*
X-linked Retinoschisis	4.1	2	*RS1 (3)*	6.2	8	*RS1 (8)*
FEVR	2.7	2	*LRP5 (2)*	3.1	4	*FZD4 (3), TSPAN12*
Cone Dystrophies	2.7	2	*IRXB, KCNV2*	0.7	1	*IRXB*

Abbreviations: LCA, Leber’s congenital amaurosis; MD, macular dystrophy; CSNB, congenital stationary night blindness; CHM, choroideremia; FEVR, familial exsudative vitreoretinopathy; %, genetically solved cases; n, genetically solved cases.

**Table 4 ijms-25-12259-t004:** Clinical diagnoses and distribution of causative mutations in genes attributed to syndromic IRDs in infantile (0–6 years old) and juvenile (7–17 years old) at time of genetic diagnosis.

	Age Group: 0–6 Years			Age Group: 7–17 Years		
Clinical Diagnosis	%	n	Associated Genes (n Cases)	%	n	Associated Genes (n Cases)
Ciliopathies (Usher syndromes excluded)	25.9	15	*CEP290 (3), c5orf43 (1), IQCB1 (1), BBS1 (2), BBS2 (1), BBS10 (2), ALMS1 (2), SLCT1 (2)*	25.4	13	*CEP290 (2), CEP120 (1), IQCB1 (1), WDR19 (1), BBS12 (1), BBS10 (4), ALMS1 (2)*
Usher syndrome type 2	5.6	3	*USH2A (3)*	11.8	6	*USH2A (6)*
Usher syndrome type 1	5.6	3	*CDH23 (2), PCDH15 (1)*	7.8	4	*CDH23 (2), CLRN1 (1)*
Stickler syndromes	14.8	8	*COL2A1 (5), COL11A1 (3)*	11.8	6	*COL2A1 (5), COL18A1 (1)*
Vitreoretinal disorders/malformations (syndromic)	14.8	8	*NDP (3), LAMA1 (1), KIF11 (1), RNU4ATAC (1), PACS2 (1), CTNNB1 (1)*	11.8	6	*KIF11 (5), KMT2D (1)*
Syndromic albinism	5.6	3	*LYST (1), HPS5 (1), HPS6 (1)*	3.9	2	*HPS6 (2)*
Macular dystrophy and cone–rod dystrophy (syndromic)	3.7	2	*CLN2 (1), GLB1 (1)*	7.8	4	*CLN3 (2), CDH3 (2)*
Rod–cone dystrophy *	3.7	2	*COH1 (1), CNNM4 (1)*	5.9	3	*COH1 (1), HGSNAT (1), CNNM4 (1)*
Cone dystrophy (syndr.)	0	0		3.9	2	*NBAS (2)*
Mitochondriopathies	3.7	2	*Pearson (1), SCO1 (1)*	7.8	4	*KSS (2), MT-TL1 (1), Pearson (1)*
Phakomatoses and others	3.6	2	*TSC1 (1), NF1 (1)*	3.9	2	*NF1(1), GRIN2A (1)*
Metabolic	9.4	5	*OAT (1), GLB1 (1), MMAHCD (1), PMM2 (1), HADHA (1)*	3.9	2	*OAT (1), HADHA (1)*

Abbreviations: %, genetically solved cases; n, genetically solved cases; * syndromic, non-ciliopathies.

**Table 5 ijms-25-12259-t005:** Clinical phenotypes and retrospective refinement of the clinical diagnosis to isolated and syndromic IRDs according to identified pathogenic variants.

	Infantile Cohort (0–6 Years)						Juvenile Cohort (7–17 Years)					
	Clinical Phenotype in Infantile Group *		Isolated IRD After Genetic Testing		Syndromic IRD After Genetic Testing		Clinical Phenotype in Juvenile Cohort *		Isolated IRD After Genetic Testing		Syndromic IRD After Genetic Testing	
	% *	n *	%	n	%	n	%*	n *	%	n	%	n
Achromatopsia	6.3	8	75.0	6	25.0	2	4.7	8	75.0	6	25	2
Albinism	12.6	6	81.3	3	18.8	3	2.9	5	60.0	3	40	2
Bardet–Biedl syndrome	3.9	5			100	5	3.5	6			100	6
Morbus Best	1.6	2	100	2			7.6	13	100	13		
Choroideremia	1.6	2	100	2			2.3	4	100	4		
CSNB	3.1	4	100	4			2.9	5	100	5		
LCA	25.2	32	75.0	24	21.9	7	7.6	13	92.3	12	7.7	1
FEVR	1.6	2	100	2			2.3	4	100	4		
Stickler syndromes	7.1	9			100	9	3.5	6			100	6
Macula dystrophy (other than Stargardt and Best)	2.4	3			100	3	7.6	13	69.2	9	30.8	4
Rod–cone dystrophy	4.7	6	66.7	4	33.3	2	7.6	13	69.2	9	30.8	4
Retinitis pigmentosa	1.6	2			100	2	12.8	22	54.5	12	45.5	10
X-linked Retinoschisis	1.6	2	150	3	0.0		4.7	8	100	8		
Stargardt’s disease	3.1	4	100	0	0		11.6	20	100	20		
Cone dystrophy	0.8	1	100	1	0.0		2.9	5	60.0	3	40.0	2
Cone–rod dystrophy	3.1	4	50.0	2	50.0	2	10.5	18	94.4	17	5.6	1
Others	8.7	11	18.2	2	81.8	9	5.2	9			100	9
Presymptomatic	11.8	15	46.7	7	53.3	8		0				

CSNB, congenital stationary night blindness; LCA, Leber’s congenital amaurosis; FEVR, familial exsudative vitreoretinopathy; *, clinical phenotype assigned prior to genetic testing; n *, total number of cases; % *, percentage of total cases of the respective age group with the respective clinical phenotype; %, percentage of cases attributed to either syndromic or isolated IRD after genetic testing in the respective age group; n, number of cases assigned either syndromic or isolated IRD after genetic testing in the respective age group; others, comprises individual clinical phenotypes with low individual prevalence including Kearns–Sayre syndrome, Norrie syndrome, *KIF11*-associated IRDs, tuberous sclerosis, gyrate atrophy, neurofibromatosis I, cb1d-MMA; CDG-1A, *SCO1*, *CTNNB1*, and *PACS2*-associated phenotypes.

## Data Availability

Data are contained within the article and Appendix A.

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
