# Peer review of "Phenotypic and Genetic Spectrum in 309 Consecutive Pediatric Patients with Inherited Retinal Disease"

_ijms, 2024, doi:10.3390/ijms252212259_

Round 1
Reviewer 1 Report
Comments and Suggestions for Authors
Children could result as unaffected in a presymptomatic phase of the diseases. A paragraph explaining the importance of deep phenotyping and electrodiagnostic testing in dominant pedigrees would further improve this interesting and well-presented work.
Author Response
Comment: Children could result as unaffected in a presymptomatic phase of the diseases. A paragraph explaining the importance of deep phenotyping and electrodiagnostic testing in dominant pedigrees would further improve this interesting and well-presented work.
Response:
We want to thank the reviewer for the important remark. We fully agree that deep clinical phenotyping is of very important not only in autosomal dominant but also in x-linked pedigrees. This is of even higher relevance in countries such as Germany, where the legal situation prohibits presymptomatic genetic testing. However, this may differ between countries. For this reason in this group of patients fundus autofluorescence and ERG may already reveal obvious preclinical signs of an IRD in clinically asymptomatic patients. These findings would then allow to proceed for genetic testing in order to confirm the diagnosis (in Germany).
In the view of new treatments to come such as the still pending gene therapies for early, rather aggressive IRDs such as RPGR-RP or choroideremia this is of high relevance. Until then however, we are back to the ethical question of the right not to know and whether such presymptomatic knowledge wouldn´t be a psychologic burden for child and parents.
The following paragraph has been added to the conclusion to emphasize this important point and to put it in context with the German background of this study in lines 615-621 as follows:
“Such deep clinical phenotyping in asymptomatic patients can be of particular relevance for subjects with a.d. or x-linked pedigrees, but also in younger siblings of children with autosomal recessive traits and encourage to proceed to molecular genetic testing. In these respect, local regulatories must carefully considered. In Germany, for example, the Genetic Diagnostics Act (§ 3 No. 8 GenDG) stipulates that predictive molecular genetic testing must not be performed on pre-symptomatic patients who are unable to give informed consent”.
Reviewer 2 Report
Comments and Suggestions for Authors
This study retrospectively analyzed the phenotypic and genetic spectrum of inherited retinal dystrophies (IRDs) in 309 pediatric patients. Patients were divided into two age groups—preschool (0-6 years) and schoolchildren (7-17 years). Preschool children often presented with nystagmus, lack of visual interest, or nyctalopia, while schoolchildren commonly exhibited declining visual acuity, nyctalopia, or high myopia. Genetic variants were identified in 96 different genes. Syndromic IRDs were more prevalent in preschool children. The study highlights age-related differences in the genotypic and phenotypic spectrum of IRDs and emphasizes the importance of early screening. The authors have provided valuable insights into the phenotypic and genetic distributions of IRDs in their genetic diagnosis service. This information is clinically relevant and beneficial. I have a few minor suggestions for the authors:
1. Please include more detailed information on the NGS panels utilized for genetic testing. It would be helpful to reference relevant studies that describe these panels.
2. A flowchart illustrating the clinical pathway of the study subjects would enhance understanding of the practice patterns at the authors’ site and may help distinguish these clinical outcomes from those at other genetic service centers.
3. A brief summary of the frequency of use for each panel would be a valuable addition.
4. Please spell out "RCD" before using the abbreviation in the text.
Comments on the Quality of English LanguageThe English writing is fine.
Author Response
We want to thank the reviewer for the careful reading of our manuscript and answer the questions as follows:
Comment 1: Please include more detailed information on the NGS panels utilized for genetic testing. It would be helpful to reference relevant studies that describe these panels. Response 1: We thank the reviewer for this very important remark. In this respect we need to explain that in the genetic diagnostic work-flow of the present work, the clinical ophthalmologists requested NGS of gene panels or Sanger sequencing of specific genes based on the clinical phenotype and using the options provided on the request forms of the three molecular genetic test services as detailed in the M&M section. The referral always included a comprehensive medical report with details on the clinical phenotypes and presumable associated systemic features as evident at time of referral for genetic testing.
Until 2018 the molecular genetic services relied on NGS of gene panels. From 2018 - 2019 all three gradually switched to whole exome sequencing (WES). Since then, virtual panels based on human phenotype ontology (HPO) annotation were bioinformatically extracted from whole exome sequencing (WES) data for core phenotyping.
During the study period the virtual panels of all three centers have constantly been adapted in agreement with the literature at least once a year as new genes were discovered. Therefore, the exact composition of the individual panels in detail not only varies over time, but also from laboratory to laboratory. The frequency of the requested panels, however, complies with the clinically suspected phenotype as listed in Tables 3 and 4 (Response to Comment 3). Ultimately, all candidate genes known at that time were read out.
In several cases the identification of causative mutations led to refinement or even reversal of the initial clinical diagnoses (see also Table 5). For example, in some cases of suspected non-syndromic albinism, Hermansky Pudlak Syndrome (HPS) genes, which are not present in the respective gene panel were also read out, which lead to the diagnosis of HPS. Also, phenotypic LCA several times was molecular genetically attributed to various subtypes of ciliopathies or X-linked retinoschisis, although the responsible genes are not part of the “LCA” panel.
For the reasons described above, it is difficult to provide references about the genes included in the panels at any given time and virtual panels applied for a given patient.
One exception is the study of Weisschuh et al (Weisschuh N, Obermaier CD, Battke F, Bernd A, Kuehlewein L, Nasser F, Zobor D, Zrenner E, Weber E, Wissinger B, Biskup S, Stingl K, Kohl S. Genetic architecture of inherited retinal degeneration in Germany: A large cohort study from a single diagnostic center over a 9-year period. Hum Mutat. 2020 Sep;41(9):1514-1527. doi: 10.1002/humu.24064. Epub 2020 Jun 29. PMID: 32531858). Molecular genetic data in this study were acquired in the partially the same recruitment period as in our study in cooperation with CeGAt. Therefore, in the samples sent to CeGaT within the same recruitment period as in the above adult cohort genetic pasting should have been based on the same NGS-panels. The study is referenced as Ref. 2.
We added some information on this background in lines 543-549 and hope that this improves the manuscript.
For the interest of the reviewer, we attach a table listing the genes included in the virtual panels as to Nov 8th 2024 according to the participating genetic services. The table illustrates that neither requestable panels (according to phenotype), nor type or number of genes included in the panels are identical. Furthermore, although the table was created today, it is most likely still inaccurate, because the genes included are constantly changing due to internal updates.
Comment 2: A flowchart illustrating the clinical pathway of the study subjects would enhance understanding of the practice patterns at the authors’ site and may help distinguish these clinical outcomes from those at other genetic service centers.
Response 2: We thank the reviewer for making the important point. To further clarify our work flow we have implemented a flow-chart illustrating the clinical pathway as Fig. 4.
Comment 3: A brief summary of the frequency of use for each panel would be a valuable addition.
Response 3: Please see answer to question 1.
Comment 4: Please spell out "RCD" before using the abbreviation in the text.
Response 4: We thank for the careful reading of the manuscript. The abbreviation now is explained before being used in the text (Abstract).
